# Attenuation of Ventilation-Enhanced Epithelial–Mesenchymal Transition through the Phosphoinositide 3-Kinase-γ in a Murine Bleomycin-Induced Acute Lung Injury Model

**DOI:** 10.3390/ijms24065538

**Published:** 2023-03-14

**Authors:** Li-Fu Li, Chung-Chieh Yu, Chih-Yu Huang, Huang-Pin Wu, Chien-Ming Chu, Ping-Chi Liu, Yung-Yang Liu

**Affiliations:** 1Division of Pulmonary and Critical Care Medicine, Department of Internal Medicine, Chang Gung Memorial Hospital, Keelung 20401, Taiwan; lfp3434@cgmh.org.tw (L.-F.L.);; 2Department of Internal Medicine, Chang Gung University, Taoyuan 33302, Taiwan; 3Department of Respiratory Therapy, Chang Gung Memorial Hospital, Keelung 20401, Taiwan; 4Community Medicine Research Center, Chang Gung Memorial Hospital, Keelung 20401, Taiwan; 5Chest Department, Taipei Veterans General Hospital, Taipei 112201, Taiwan; 6School of Medicine, Faculty of Medicine, National Yang Ming Chiao Tung University, Taipei 112304, Taiwan; 7Institute of Clinical Medicine, School of Medicine, National Yang Ming Chiao Tung University, Taipei 112304, Taiwan

**Keywords:** acute lung injury, apoptosis, epithelial–mesenchymal transition, phosphoinositide 3-kinase-γ, ventilator-induced lung injury

## Abstract

Mechanical ventilation (MV) used in patients with acute lung injury (ALI) induces lung inflammation and causes fibroblast proliferation and excessive collagen deposition—a process termed epithelial–mesenchymal transition (EMT). Phosphoinositide 3-kinase-γ (PI3K-γ) is crucial in modulating EMT during the reparative phase of ALI; however, the mechanisms regulating the interactions among MV, EMT, and PI3K-γ remain unclear. We hypothesized that MV with or without bleomycin treatment would increase EMT through the PI3K-γ pathway. C57BL/6 mice, either wild-type or PI3K-γ-deficient, were exposed to 6 or 30 mL/kg MV for 5 h after receiving 5 mg/kg AS605240 intraperitoneally 5 days after bleomycin administration. We found that, after bleomycin exposure in wild-type mice, high-tidal-volume MV induced substantial increases in inflammatory cytokine production, oxidative loads, Masson’s trichrome staining level, positive staining of α-smooth muscle actin, PI3K-γ expression, and bronchial epithelial apoptosis (*p <* 0.05). Decreased respiratory function, antioxidants, and staining of the epithelial marker Zonula occludens-1 were also observed (*p <* 0.05). MV-augmented bleomycin-induced pulmonary fibrogenesis and epithelial apoptosis were attenuated in PI3K-γ-deficient mice, and we found pharmacological inhibition of PI3K-γ activity through AS605240 (*p <* 0.05). Our data suggest that MV augmented EMT after bleomycin-induced ALI, partially through the PI3K-γ pathway. Therapy targeting PI3K-γ may ameliorate MV-associated EMT.

## 1. Introduction

Acute respiratory distress syndrome (ARDS), a severely debilitating disease, can lead to death associated with multiple organ failure [1,2]. Mechanical ventilation (MV) is often used to maintain life in patients with ARDS. However, despite the widespread use of lung-protective ventilation strategies, ventilator-induced lung injury (VILI), which is characterized by an initial inhomogeneous inflammatory reaction or epithelial injury followed by a fibroproliferative phase with extracellular matrix (ECM) accumulation, continues to occur [1,2,3,4]. 

The three major factors that drive the differentiation of fibroblasts to myofibroblasts are high tidal volume mechanical stress, local increase in transforming growth factor-β1 (TGF-β1), and the presence of the extra domain A splice variant of fibronectin [1,2]. The lesional effect of MV on the ECM may be influenced by the following factors: increased transpulmonary pressure, heterogeneous distribution of ventilation, increased tissue stretch, and decreased pulmonary lymphatic drainage [1,4,5]. However, the conditions that determine alveolar recovery and fibrosis remain unclear.

MV may increase pulmonary oxidative stress and is a potent stimulus for the production of TGF-β1 and plasminogen activator inhibitor-1 (PAI-1), thereby, damaging pulmonary epithelial cells [1,2,6]. As a key profibrogenic cytokine, TGF-β1 plays a crucial role in the transdifferentiation of epithelial cells into myofibroblast-like cells—a phenomenon known as epithelial–mesenchymal transition (EMT) [1,2]. PAI-1 has been found to be involved in fibrinolytic defects associated with various forms of lung injuries [6,7]. EMT is characterized by the loss of epithelial markers (i.e., E-cadherin and Zonula occludens-1 (ZO)-1) and apical–basal polarity and by cytoskeletal rearrangement, transition to a spindle-shaped morphology, and the accumulation of mesenchymal markers (i.e., α-smooth muscle actin (α-SMA) and N-cadherin) [1,2].

Although EMT is a crucial downstream mechanism of most fibrosing diseases, whether it signifies a leading process in the development of idiopathic pulmonary fibrosis has been widely discussed and remains a matter of debate [8]. Previous studies have demonstrated reduced mitochondrial respiration and the release of reactive oxygen species (ROS) in the lung tissue of patients with pulmonary fibrosis [9,10]. Using rat lungs and primary rat alveolar epithelial cells (AECs) exposed to normoxia or hypoxia, the endoplasmic reticulum (ER) stress and hypoxia-inducible factor (HIF) signaling pathways were demonstrated to regulate hypoxia and intracellular calcium involved in EMT induction of AECs [11].

An increasing body of evidence shows that the use of MV in patients is frequently associated with the risk of ventilator-associated pneumonia and lung fibrosis with loss of volume [12]. However, the effective therapeutic approaches for pulmonary fibrosis treatment are still lacking. Therefore, exploring potential molecular mechanisms underlying the transition from acute lung inflammation to pulmonary fibrosis is crucial to facilitating the search for new molecular markers and promising therapeutic targets as well as the transition of patients with ALI away from long-term ventilatory support [3].

Phosphoinositide 3-kinase (PI3K) is a cellular lipid kinase that phosphorylates the 3-hydroxyl of the phosphatidylinositol ring to generate the lipid second messenger phosphatidylinositol 3,4,5-triphosphate [13,14,15]. Class I PI3Ks, containing the class IA (p110α, β, and δ) and class IB (p110γ) isoforms, have been the focus of recent studies [13]. Class IA kinases, which form complexes with regulatory p85-related subunits containing SH2, are primarily activated through receptor tyrosine kinases, whereas class IB p110γ, which forms complexes with regulatory subunit p101 or p84, is stimulated by G protein-coupled receptors [13].

PI3K-γ is expressed on leukocytes and on other cell types, including endothelial cells and fibroblasts, and may regulate the influx of leukocytes into the lungs [13,14,15]. In the absence of PI3K-γ, decreased influx and activation of leukocytes, reduced transcription of fibrogenic markers, and reduced angiogenesis in the lung tissue were observed [13]. The serine/threonine kinase-protein kinase B (Akt) pathway is activated in pulmonary fibrosis, and the inhibition of this pathway reduces fibroblast to myofibroblast differentiation and ameliorates pulmonary fibrosis [13,14,16].

The PI3K and its downstream Akt pathway may regulate pulmonary fibrosis by inducing the expression of vascular endothelial growth factor (VEGF) [13,14]. VEGF was shown to be dysregulated in patients with pulmonary fibrosis and in an animal model of bleomycin-induced lung fibrosis [14,17]. However, the underlying mechanisms of the PI3K-γ pathway in mechanical stretch-induced ALI and fibrogenesis are not fully elucidated.

We used our mouse model of VILI after bleomycin treatment in this study [1,2]. The goals of this study were as follows: (1) evaluating PI3K-γ expression associated with the development of pulmonary fibrogenesis during MV; (2) comparing the oxidative load and inflammatory cytokine production between MV and bleomycin-mediated lung damage; (3) examining the role of PI3K-γ in the VILI through PI3K-γ homozygous knockout and pharmacologic inhibition through AS605240, a PI3K-γ inhibitor [18]; and (4) examining the role of the PI3K-γ signaling pathway in bronchial epithelial apoptosis using bleomycin. We hypothesized that MV with or without bleomycin treatment would increase lung inflammation, EMT, pulmonary fibrogenesis, the production of free radicals, and epithelial apoptosis through upregulating the PI3K-γ pathway.

## 2. Results

### 2.1. Inhibition of Bleomycin-Stimulated MV-Induced VILI through AS605240

We used high-tidal-volume (V_T_ 30 mL/kg) and a low-tidal-volume (V_T_ 6 mL/kg) MV with room air for 5 h to induce VILI in mice. We examined the injurious effects of overdistension and treatment effects of intraperitoneally delivered AS605240. The physiological conditions at the beginning and end of MV are shown in Appendix A. The normovolemic statuses of the mice were maintained by monitoring their mean artery pressure.

The deleterious effects of MV-induced changes in microvascular permeability, lung water content, hypoxemia, and impaired respiratory function were identified by measuring the level of Evans blue dye (EBD) in the lung (Figure 1A), lung wet-to-dry weight ratio (Figure 1B), level of gas exchange (i.e., the partial pressure of oxygen (PaO_2_)/fraction of inspired oxygen (FiO_2_), Figure 1C), and respiratory function enhanced pause (Penh) (Figure 1D). Furthermore, the oxidant load and inflammatory cytokine levels were analyzed to determine the level of oxidative stress and the amount of profibrogenic cytokines in fibroblasts in VILI (Figure 2).

Increased EBD, wet-to-dry weight ratio, malondialdehyde (MDA, a maker of lipid peroxidation), PAI-1, and VEGF protein production were observed in mice treated with bleomycin subjected to high-tidal-volume MV compared with the other MV treatment groups and the nonventilated control mice (Figure 1 and Figure 2). The exacerbation of gas exchange and lung inflammation in mice administered high-tidal-volume MV and treated with bleomycin were substantially ameliorated after the administration of AS605240 (Figure 1 and Figure 2).

### 2.2. Reduction in the Effects of MV on Bleomycin-Enhanced Collagen Fiber Production and Fibrogenic Markers through AS605240

Masson’s trichrome staining (Figure 3A, Appendix A) and transmission electron microscopy (TEM) were used to determine the effects of MV on the ultrastructure of the accumulated peribronchiolar and parenchymal collagen fibers. Collagen fibers were more prevalent in the ECM in the mice treated with bleomycin and subject to a VT of 30 mL/kg compared with the other MV-treated and the nonventilated control mice (Figure 3A,B). In addition, we employed micro-computer tomography (CT), a primary imaging technique where lung fibrosis is monitored clinically, to examine the damaged lung structure [19].

Pulmonary fibrotic changes, including increased reticular opacities, honeycombing, traction bronchiectasis, and ground glass opacities in the lower lungs, were aggravated in the mice ventilated at a VT of 30 mL/kg and treated with bleomycin compared with the other MV treatment groups and the nonventilated control mice as measured using a densitometry of Hounsfield Unit (HU) of lung parenchyma (Figure 3C). Furthermore, the expressions of ZO-1 (i.e., an epithelial marker) and α-SMA (i.e., a mesenchymal marker) were measured using immunofluorescent staining to identify the transition of cell types involved in the lung-stretch-induced EMT (Figure 4A). In addition, fibrosis scoring using Masson’s trichrome staining was calculated to quantify the effects of MV on ECM deposition (Figure 4B).

The effects of MV on the increase in collagen fibers, downregulation of ZO-1, and upregulation of α-SMA were substantially diminished through pharmacologic inhibition with AS605240 (Figure 4).

### 2.3. Suppression of the Effects of MV on Bleomycin-Stimulated PI3K-γ Protein Expression through AS605240

As PI3K-γ activation modulates stretch-augmented fibrosis, we assessed the PI3K-γ expression to examine the role of the PI3K-γ signaling pathway in VILI (Figure 5) [13,14,15]. Western blot analyses showed increased PI3K-γ expression in the mice treated with bleomycin and MV with room air compared with the other MV treatment groups and the nonventilated control mice (Figure 5A). Furthermore, the increase in PI3K-γ expression in mice treated with bleomycin and high-tidal-volume MV was substantially attenuated through the inhibition of AS605240 (Figure 5A).

Immunohistochemistry was used to further define the cell types involved in the lung stretch-induced fibrogenesis and to verify the effects of AS605240 on PI3K-γ activation in VILI (Figure 5B,C). In positive immunohistochemical staining results for PI3K-γ, substantially increased levels were noted in the airway epithelial cells in the mice treated with bleomycin and high-tidal-volume MV compared with the other MV treatment groups and the nonventilated control mice (Figure 5B,C). Consistent with the results of the Western blot, the administration of the PI3K-γ inhibitor, AS605240, prevented the VT30-induced activation of PI3K-γ (Figure 5B,C).

### 2.4. Inhibition of Bleomycin-Stimulated MV-Induced Lung Inflammation and EMT in PI3K-γ-Deficient Mice

PI3K-γ-deficient mice were used to determine the role of PI3K-γ in stretch-induced lung injury—specifically whether the improvements in lung injuries and fibrogenesis after AS605240 administration were induced through PI3K-γ expression (Figure 6, Figure 7, Appendix A). The effects of MV on various parameters, including changes in the microvascular permeability and pulmonary edema, hypoxemia, lung function, inflammatory cytokine generation, oxidative stress, accumulation of collagen fibers, fibrogenic markers, and fibrosis scoring, were substantially decreased in the PI3K-γ-deficient mice treated with bleomycin and high-tidal-volume MV (*p* < 0.05; Figure 6, Figure 7, Appendix A). In the course of ARDS, lung inflammation and fibrogenesis interact to some degree [2].

We observed that neutrophil counts in the BAL fluid increased in mice subjected to VT at 30 mL/kg compared with those subjected to VT at 6 mL/kg and the control mice. Significantly, PI3K-γ knockout and pharmacologic inhibition with AS605240 substantially reduced neutrophil infiltration (neutrophil counts: nonventilated control wild-type mice without bleomycin = 1.0 ± 0.1, nonventilated control wild-type mice with bleomycin = 2.1 ± 0.1, VT 6 mL/kg wild-type mice with bleomycin = 5.7 ± 0.3 *, VT 30 mL/kg wild-type mice without bleomycin = 24.1 ± 1.8 *, VT 30 mL/kg wild-type mice with bleomycin = 42.5 ± 2.6 *, VT 30 mL/kg PI3K-deficient mice with bleomycin = 20.7 ± 1.4 *, and VT 30 mL/kg wild-type mice after AS605240 with bleomycin = 28.3 ± 2.9 * ×104/mL BAL; * *p* < 0.05 versus control (Appendix A)).

### 2.5. Reduction of the Effects of MV on Bleomycin-Enhanced Expression of Caspase-3 and Epithelial Apoptosis in PI3K-γ-Deficient Mice

In addition to its role in oxidative stress, caspase-3 is central to the intrinsic apoptotic pathway [20]. Capase-3 expression and TEM were conducted to explore the functions of the caspase-3 pathway and apoptosis of airway epithelial cells in bleomycin-associated VILI (Figure 8). The substantial increase in the expression of cleaved caspase-3 (active form) but decrease in the expression of non-cleaved caspase-3 was observed in the mice treated with bleomycin and high-tidal-volume MV compared with the other MV treatment groups and the nonventilated control mice (Figure 8A).

Epithelial apoptosis was characterized by characteristic nuclear condensation of the bronchial epithelium in the mice treated with MV and bleomycin (Figure 7B). Specifically, MV- and bleomycin-stimulated caspase-3 activity and apoptosis in the airway epithelia were lower in the PI3K-γ-deficient mice than in control mice (*p* < 0.05; Figure 8). As no statistically significant differences were observed between the wild-type and PI3k-γ-deficient nonventilated control mice or wild-type and AS605240 treatment nonventilated control mice, both with and without bleomycin, the data are not presented in this paper. Our results indicate that the inhibition of the PI3K-γ pathway leads to the suppression of MV- and bleomycin-induced inflammatory processes and pulmonary fibrogenesis (Figure 9).

## 3. Discussion

Pathological fibrogenesis after an episode of ALI is often a result of abnormal wound healing, characterized by increased lung inflammation, severe hypoxemia, and the loss of lung compliance [17]. Early fibroproliferation occurs within 24 h of the onset of ARDS [4,21,22]. Fibroblast proliferation and ECM accumulation begin 4–14 days after bleomycin challenge [23,24]. Furthermore, Tager et al. demonstrated that the angiogenic and fibroblast chemotactic activities generated in mouse lungs 5 days after a bolus of bleomycin administration was substantially up-regulated [24].

Further, we have shown that there were significant increases of fibroblast accumulation and fibrogenic markers in this animal model using MV 5 days after bleomycin administration [1,2]. Five days of bleomycin administration was thus used in our study to focus on the major target involved in the early phase of pulmonary fibrosis after ALI. Survivors of ARDS may develop permanent pulmonary fibrosis, resulting in decreased quality of life and dependence on MV [21,25].

Therefore, early intervention aimed at restoring fibroproliferative activity to prevent subsequent pulmonary fibrogenesis progression may improve clinical outcomes in patients with ARDS. The parameters determining alveolar recovery and advanced fibrogenesis have not been fully explored. A deeper understanding of the exact mechanism of altered ECM deposition in the pathogenesis VILI is crucial for developing potential strategies to reduce the long-term use of MV and associated intensive care unit mortality.

To investigate the mechanisms of pulmonary fibrogenesis in mice treated with MV, mice were subjected to MV for 5 h and were administered with 5 days of bleomycin to focus on major targets included in the early stages of fibroproliferation after ALI. MV is crucial to maintain adequate ventilation and gas exchange in patients with ARDS; however, MV can trigger VILI or aggravate the inflammatory response to fibrogenesis, leading to VILI-associated lung fibrosis. Though ARDS network’s clinical trials have revealed that low-tidal-volume MV is safer than high-tidal-volume MV, the mortality of patients with ARDS has remained substantially high. The mechanisms of MV-associated lung fibrosis or post-ARDS pulmonary fibrogenesis need to be further investigated [5,26,27].

In this study, we demonstrated that extensive epithelial injury results in increased alveolar capillary permeability and inflammatory parameters in our two-hit animal model. MV-augmented bleomycin-induced pulmonary fibrogenesis and epithelial apoptosis were attenuated in PI3K-γ-deficient mice and pharmacological inhibition of PI3K-γ activity through AS605240. Moreover, using micro-CT to evaluate changes in pulmonary fibrogenesis through densitometry can further provide quantitative data for the comparison of distinct groups [19]. A previous clinical study reported that the CT assessment of lung fibroproliferation can serve as a prognostic indicator of a patient’s MV dependence and functional outcomes [22]. In addition, we examined the role of PI3K-γ in regulating the pathogenesis in the fibroproliferation of VILI.

Patients with ALI with greater fibrotic changes required prolonged use of MV [5,21]. The proinflammatory and profibrotic responses in patients with ALI may become uninhibited and can induce fibrogenesis with a subsequent decline in lung function and prolonged use of MV [21,25]. EMT is a tissue repair process after injury, in which alveolar epithelial cells lose epithelial features while acquiring mesenchymal cell characteristics and are reprogrammed with alterations in cell polarity, morphological transformation, secretory phenotype, intercellular junctions, and cytoskeletal rearrangements [5,28].

Alveolar epithelial cells are mechanically stretched during respiration, whereas capillary endothelial cells are mainly affected by hydrostatic pressure, shear stress, and strain [4,5]. ALI is characterized by impaired gas exchange of alveoli and airways, which includes terminal and respiratory bronchioles. Therefore, we explored high-tidal-volume MV-induced peribronchiolar and parenchymal collagen accumulation in the murine bronchial epithelium in addition to the alveolar injury found in our (Figure 2A and Figure 7A) and others’ studies in mice [29].

The major distinct cellular mechanisms driving the differentiation of fibroblasts to myofibroblasts include the activation of stretch-sensitive ion channels in pulmonary epithelial and endothelial cells, the disruption of cell plasma membranes, and the loss of tight junction structure and cell–cell attachment associated with disturbances in the degradation of occluding and actin perturbations [4,21,25]. The activation of EMT can cause the massive production and deposition of ECM, thereby, leading to lung remodeling and subsequent pathological fibrogenesis [30].

High-tidal-volume MV can damage alveolar epithelial cells and promote the transdifferentiation of epithelial cells into myofibroblasts, which then results in abnormal wound repair and healing through the EMT signaling pathway [21,31,32]. In an animal model of ventilator-induced pulmonary fibrosis, the sustained activation of M2 macrophages induced EMT to polarize epithelial cells into a profibrotic direction through the activation of TGF-β1 signaling [33].

In this study, we demonstrated that high-tidal-volume MV and bleomycin treatment in mice augmented the EMT process as indicated by a decrease in local epithelial marker (i.e., ZO-1) and an increase in a local mesenchymal marker (i.e., α-SMA). The activation of the EMT pathway may contribute to ventilation-augmented bleomycin-induced pulmonary fibrogenesis. Therefore, investigating the regulation of EMT is crucial in preventing the development of VILI-associated pulmonary fibrogenesis. Except for the TGF-β1-dependent pathway, other inflammatory mediators may be involved in the development of pulmonary fibrogenesis in a TGF-β1-independent manner [13,14].

MV can regulate gene expression and structural remodeling of the ECM by increasing the transpulmonary pressure, changing the ventilation distribution, increasing tissue stretch, decreasing pulmonary lymphatic drainage, and directly secreting various growth factors, including PAI-1 and VEGF. The coagulation cascade and angiogenesis are involved in the pathogenesis of pulmonary fibrosis at the initial stage as key driving forces promoting the progression of pulmonary fibrosis [30].

The EMT of epithelial and endothelial cells is affected by the coagulation cascade and angiogenesis-related cytokines (e.g., tissue factor, PAI-1, and VEGF) [13,14]. PAI-1 rapidly inhibits both the tissue-type plasminogen activator and the urokinase-type plasminogen activator [13]. Moreover, PAI-1 is crucial for the differentiation of fibroblasts to myofibroblasts, inhibition of airspace fibrinolysis, and promotion of alveolar epithelial cell apoptosis, which can contribute to the loss of epithelial regenerative ability and progression to irreversible pulmonary fibrosis [7,30]. In mice, the homozygous deletion of PAI-1 mitigates bleomycin-induced pulmonary fibrogenesis, whereas the increased expression of PAI-1 augments pulmonary fibrogenesis [1,30].

In addition, neovascularization is required for fibroblast proliferation and ECM deposition during repair and plays a critical role in lung fibrogenesis [15]. Patients with IPF had decreased levels of endothelial progenitor cells and increased levels of compensatory proangiogenic mediator VEGF, a key cytokine of angiogenesis, which may contribute to the growth of alveolar epithelial type II cells and the production of surfactant, angiogenesis, and anti-apoptotic effects [13,34].

VEGF levels have been found to correlate with the CT scores of fibrosis in patients with IPF, and lung fibrosis can be attenuated by inhibiting VEGF expression through downregulating PI3K [13,14,17]. In this study, we demonstrated that high-tidal-volume MV aggravated PAI-1 and VEGF production after bleomycin exposure. These findings suggest that MV can produce and exacerbate the generation of coagulation cascade and angiogenesis, thus, providing a pathogenic microenvironment for the promotion of pulmonary fibrosis, which is consistent with the results of previous studies [13,15,34].

Although pirfenidone and nintedanib are approved for the treatment of IPF, their effects in slowing the progression of lung fibrogenesis and decline in pulmonary function are limited, and both have undesired side effects [1,14]. Exploring more molecular mechanisms involved in the reparative phase of ALI and determining optimal treatments is required to ameliorate EMT and lung fibrogenesis. PI3K represents a crucial signaling pathway during oxidative stress and fibrogenesis [14,35].

During ALI, PI3K-γ modulates leukocyte activation and influx into the lung, which results in the generation of ROS and reduction of antioxidant ability [14]. Notably, PI3K-γ overexpression has been demonstrated in the lung tissues of patients with IPF and fibroblasts [14,36]. The activation of PI3K is related to the overexpression of α-SMA in the transition of human lung fibroblasts to myofibroblasts, and the inhibition of PI3K-γ can suppress α-SMA expression in IPF fibroblasts [14,36]. Previous studies have found that PI3K can promote EMT and contribute to the pathogenesis of bleomycin-induced pulmonary fibrosis, suggesting that PI3K plays a pivotal role in lung fibrogenesis by regulating EMT [4,13,14].

In addition, PI3K-γ plays a key role in angiogenesis, which is required for lung repair, and the inhibition of PI3K attenuates pulmonary fibrosis through suppressing VEGF activation [4,34]. Previous studies have revealed that PI3Ks modulate acute MV-induced vascular hyperpermeability, that the lack of PI3K-γ by gene knockout can prevent VILI, and that specific pharmacological inhibition of PI3K-γ kinase activity ameliorates VILI through reducing nitric oxide release and increasing cAMP production [13,14,37]. Furthermore, PI3K-γ knockout mice exhibited diminished pulmonary collagen fiber deposition and reduced gene expression of profibrogenic and proangiogenic mediators, suggesting that PI3K-γ deficiency protects against bleomycin-induced lung inflammation and fibrogenesis [38].

However, the detrimental role of PI3K-γ in VILI-associated lung fibrosis remains unknown. In this study, we demonstrated that high-tidal-volume MV augmented PI3K-γ activity and increased oxidative stress, vascular leakage, VEGF and PAI-1 production, and EMT activation after bleomycin exposure. Thus, the inhibition of PI3K-γ activity through a PI3K-γ specific inhibitor or a PI3K-γ gene knockout resulted in the suppression of bleomycin-stimulated MV-induced pulmonary fibrogenesis. According to our findings, the profibrogenic effects of the PI3K-γ pathway provide a promising therapeutic target for the treatment of VILI-associated pulmonary fibrogenesis through suppressing PI3K-γ signaling.

In the current study, bronchial epithelial cells were our main focus to investigate. Immunohistochemistry was used to further define the cell types involved in the lung stretch-induced fibrogenesis and to verify the effects of AS605240 on PI3K-γ activation in VILI (Figure 5B,C). Previous in vitro study demonstrated that primary human bronchial epithelial cells, in addition to type 2 alveolar epithelial cells, stimulated by TGF-β1, can initiate EMT process through a Smad 2/3-mediated pathway [39]. Furthermore, partial inhibition of the increase in EMT and fibroblast accumulation in PI3K-γ deficient mice suggested that PI3K-γ signaling was only one of the many pathways contributing to the pathogenesis of fibrogenesis associated with VILI [4,13,40].

Yes-associated protein 1 (YAP) has been revealed to modulate mechanical tension-induced alveolar regeneration, and activation of YAP restores lung fibrogenesis through the NF-kB pathway [4,40]. Platelet-derived growth factor can also promote fibroblast proliferation and activate NF-kB through the Src/PI3K/Akt pathway [13,37]. Histone modification has been demonstrated to be involved in the higher-order chromatin structure alterations in cellular apoptosis [41]. Although our pathologic findings of epithelial apoptosis revealed characteristic nuclear condensation and fragmented heterochromatin of epithelial cells, different cellular responses, including senescence, have been associated with chromatin changes [42].

Super-resolution microscopy (e.g., stochastic optical reconstruction microscopy (STORM)) may help to detect the epigenetic marks to delineate the pattern of histone medication [43]. Zhang R et al. demonstrated that MV can induce lung fibrotic changes in an animal model of ARDS-associated lung fibrosis as demonstrated by enhanced levels of hydroxyproline in lung tissues [44]. Hydroxyproline, a crucial component of the protein collagen as an early biochemical marker contributing to collagen synthesis, can serve as a supportive assay in addition to the Masson’s trichrome staining used in our study. Further studies are necessary to identify potential regulators of MV-associated pulmonary fibrogenesis.

## 4. Materials and Methods

### 4.1. Ethics of Experimental Animals

Wild-type or PI3K-γ-deficient C57BL/6 mice, weighing between 20 and 25 g and aged between 6 and 8 weeks, were obtained from Jackson Laboratories (catalog number 024587, Bar Harbor, ME, USA) and the National Laboratory Animal Center (Taipei, Taiwan) [38,45]. Briefly, homozygotes mutants (PI3K-γ-/-) exhibit an impaired neutrophil chemotaxis and respiratory burst in response to formyl peptide N-formyl-Met-Leu-Phe and C5a induction as well as reduced thymocyte survival and activation of mature T lymphocytes [38,45]. 

The lower expressions of the PI3K-γ protein in PI3K-γ-/- mice were confirmed by using a Western blot analysis. The study was performed in strict accordance with the recommendations in the Guide for the Care and Use of Laboratory Animals of the National Institutes of Health. The protocol was approved by the Institutional Animal Care and Use Committee of Chang Gung Memorial Hospital (Permit number: 2020111701). All surgery was performed under ketamine and xylazine anesthesia, and all efforts were made to minimize suffering.

### 4.2. Bleomycin Administration

Bleomycin, which acts by preventing the incorporation of thymidine into the DNA, promotes EMT by inducing DNA strand breaks [1,2]. The mice received a single dosage of 0.075 units of bleomycin in 100 μL of sterile normal saline (2 mg/kg, Sigma, St. Louis, MO, USA) intratracheally for 5 days. Bleomycin exposure results in an acute inflammatory reaction followed by pulmonary fibrosis that slowly resolves [1,2].

### 4.3. Pharmacological Inhibitors

PI3K-γ inhibitor (AS605240, Sigma, St. Louis, MO, USA) 5 mg/kg was given intraperitoneally 1 h before MV based on our present and previous studies that showed that 5 mg/kg inhibited PI3K-γ activity [18].

### 4.4. Experimental Groups

Animals were randomly distributed into nine groups in each experiment: group 1, nonventilated control wild-type mice with normal saline; group 2, nonventilated control wild-type mice with bleomycin; group 3, VT 6 mL/kg wild-type mice with bleomycin; group 4, VT 30 mL/kg wild-type mice with normal saline; group 5, VT 30 mL/kg wild-type mice with bleomycin; group 6, VT 30 mL/kg Pi3K-γ-/- mice with bleomycin; group 7, VT 30 mL/kg wild-type mice after AS605240 (5 mg/kg) administration with bleomycin; group 8, nonventilated control wild-type mice after AS605240 administration without bleomycin; and group 9, nonventilated control PI3K-deficient mice with bleomycin. In each group, three mice underwent TEM and micro-CT, and five mice underwent measurement for immunohistochemistry and immunofluorescent assay, inflammatory cytokines, oxidative and antioxidative loads, Masson’s trichrome staining, and Western blotting.

### 4.5. Measurement of Inflammatory Cytokines

PAI-1 (0.02 ng/mL) and VEGF (1.8 pg/mL) were detected in serum and bronchoalveolar lavage fluid using immunoassay kits containing primary polyclonal anti-mouse antibodies that were cross-reactive with rat and mouse PAI-1 and VEGF (VEGF: Biosource International, Camarillo, CA, USA; PAI-1: Molecular Innovations Inc., Plymouth Meeting, PA, USA). Each sample was run in duplicate according to the manufacturer’s instructions.

### 4.6. Measurement of Oxidative Stress and Antioxidant Enzyme Expression

The lungs were homogenized in phosphate buffered saline. The malondialdehyde and total antioxidant capacity in the protein extracts were measured using the Oxiselect TBARS assay kit containing thiobarbituric acid reactive substances and Oxiselect total antioxidant capacity assay kit containing uric acid (Cell Biolabs, San Diego, CA, USA). Each sample was run in duplicate and expressed as μmole/g protein according to the manufacturer’s instructions.

### 4.7. Immunoblot Analysis

The lungs were homogenized in 0.5 mL of lysis buffer as previously described [1,2]. Crude cell lysates were matched for protein concentration, resolved on a 10% bis-acrylamide gel, and electrotransferred to Immobilon-P membranes (Millipore Corp., Bedford, MA, USA). For the assay of PI3K-γ and glyceraldehyde-3-phosphate dehydrogenase, Western blot analyses were performed with the respective antibodies (New England BioLabs, Beverly, MA, USA, Santa Cruz Biotechnology, Santa Cruz, CA, USA, and Novus Biologicals, Littleton, CO, USA). Blots were developed using enhanced chemiluminescence (NEN Life Science Products, Boston, MA, USA).

### 4.8. Immunofluorescence Labeling

The lung tissues were paraffin embedded, sliced at 4 μm, deparaffinized, and stained according to the manufacturer’s instructions for an immunohistochemical kit (Santa Cruz Biotechnology, Santa Cruz, CA, USA). Lung sections were incubated with primary rabbit anti-mouse antibodies of ZO-1 and α-SMA (1:100; New England BioLabs, Beverly, MA, USA) and fluorescent secondary antibodies of Cy3-conjugated anti-rabbit (ZO-1 and α-SMA) (1:1000; Santa Cruz Biotechnology, Santa Cruz, CA, USA). Nuclear staining was performed using Hoechst solution (0.5 μg/mL; Sigma, St. Louis, MO, USA). The fluorescence-labeled slides were subsequently examined using a Leica TCS 4D confocal laser scanning microscopy system (Leica, Wetzlar, Germany).

### 4.9. Immunohistochemistry

The lungs were paraffin embedded, sliced at 4 μm, deparaffinized, antigen unmasked in 10 mM sodium citrate (pH 6.0), and incubated with rabbit PI3K-γ primary antibody (1:100; Santa Cruz Biotechnology, Santa Cruz, CA, USA) and biotinylated goat anti-rabbit secondary antibody (1:100) according to the manufacturer’s instructions for an immunohistochemical kit (Santa Cruz Biotechnology, Santa Cruz, CA, USA).

### 4.10. Masson’s Trichrome Stain and Fibrosis Scoring

The lung tissues from control, nonventilated mice, and mice exposed to high or low tidal volume MV for 5 h while breathing room air were paraffin embedded, sliced at 4 μm, deparaffinized, and stained sequentially with Weigert’s iron hematoxylin solution, Biebrish scarlet-acid fuchsin solution, and aniline blue solution according to the manufacturer’s instructions for a trichrome kit (Sigma, St. Louis, MO, USA). A blue signal indicated positive staining of collagen.

The fibrotic grade of each lung field was assessed using the criteria of Ashcroft, ranging from grade 0 to 5 as follows: grade 0: normal lung; grade 1: minimal fibrous thickening of alveolar or bronchial walls; grade 2: moderate thickening of the walls without obvious damage to the lung architecture; grade 3: increased fibrosis with definite damage to the lung structure and formation of fibrous bands or small fibrous masses; grade 4: severe distortion of structure and large fibrous areas (honeycomb lung); and grade 5: total fibrous obliteration in the field. Average number of 10 nonoverlapping fields in Masson’s trichrome staining of paraffin lung sections, six mice per group, was analyzed for each section by a single investigator blinded to the mouse genotype [1,2].

### 4.11. Micro-Computer Tomography

At the end of the study period before sacrificing, the mice were scanned using a hybrid micro-CT imaging system (nanoScan SPECT/CT, Mediso, Hungary) [46]. The micro-CT data were acquired using a high-resolution frame as the system setup, semicircular scan, tube voltage of 70 keV, 0.189 mAs, pitch of 1.0, and 720 projections. The micro-CT images with a volume size of 40 × 40 × 27 mm and an isotropic voxel size of 39 μm were reconstructed using Filtered Back Projection reconstruction with a Cosine filter. The cubic 2 mm of the region of volume (VOIs) in lung parenchyma were selected, and the HUs of each VOI were calculated and analyzed using PMOD version 4.0 image analysis software (PMOD Technologies Ltd., Zurich, Switzerland).

### 4.12. Analysis of Data

The Western blots were quantitated using an NIH image analyzer Image J 1.27z (National Institutes of Health, Bethesda, MD, USA) and presented as arbitrary units. Values are expressed as the mean ± SD from at least five separate experiments. The data of MDA, total antioxidant capacity, histopathologic assay, and oxygenation were analyzed using Statview 5.0 (Abascus Concepts, Cary, NC, USA; SAS Institute). All results of Western blots were normalized to the nonventilated control wild-type mice with room air. ANOVA was used to assess the statistical significance of the differences followed by multiple comparisons with a Scheffe’s test, and a *p* value < 0.05 was considered statistically significant.

EBD analysis, analysis of lung water, TEM, ventilator protocol, and whole body plethysmography were performed as previously described [1,2,20].

## 5. Conclusions

We demonstrated that the use of PI3K-γ specific inhibitors or PI3K-γ gene knockout can attenuate bleomycin-stimulated MV-induced pulmonary fibrogenesis by decreasing lung microvascular leakage, oxidative stress, PAI-1 and VEGF levels, microscopic and ultrastructural pathological alterations, collagen fiber staining and fibrosis scores, and CT radiodensity and by improving the oxygenation index and pulmonary mechanics through the suppression of EMT.

Our PI3K-γ-targeting therapy achieved radiological, pathological, and functional recovery in our animal model simulating lung fibrosis after ALI in clinical practice. These beneficial effects are partly caused by the blockade of the PI3K-γ signaling pathway. A deeper understanding of the precise mechanisms underlying alterations in the pathogenesis of VILI-associated lung fibrogenesis is crucial for developing potential strategies to reduce the prolonged use of MV and the morbidity and mortality of MV.

## Figures and Tables

**Figure 1 ijms-24-05538-f001:**
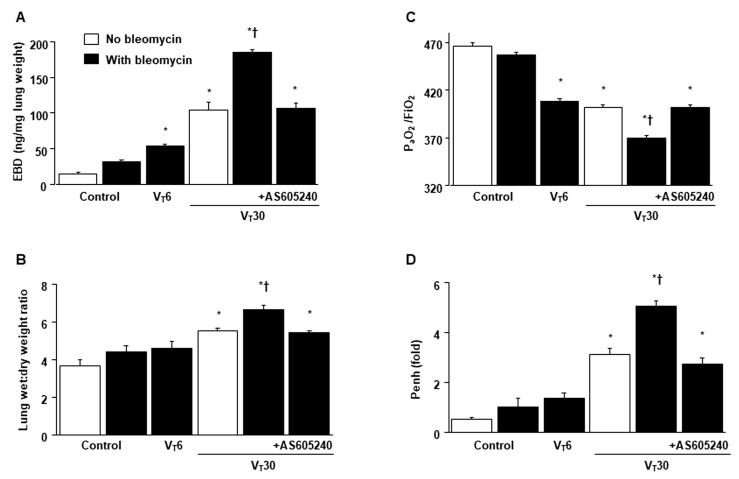
Suppression of lung stretch-induced microvascular leak, lung edema, hypoxemia, and impaired respiratory function by AS605240. Five days after administering bleomycin, (**A**) Evans blue dye analysis, (**B**) lung wet-to-dry-weight ratio, (**C**) PaO_2_/FiO_2_, and (**D**) enhanced pause from the lungs of nonventilated control mice and mice ventilated at a tidal volume of 6 mL/kg (V_T_ 6) or 30 mL/kg (V_T_ 30) for 5 h with room air (*n* = 5 per group). AS605240 5 mg/kg was given intraperitoneally 1 h before ventilation. * *p* < 0.05 versus the nonventilated control mice with bleomycin pretreatment; † *p* < 0.05 versus all other groups. EBD = Evans blue dye; FiO_2_ = fraction of inspired oxygen; PaO_2_ = partial pressure of oxygen; and Penh = enhanced pause.

**Figure 2 ijms-24-05538-f002:**
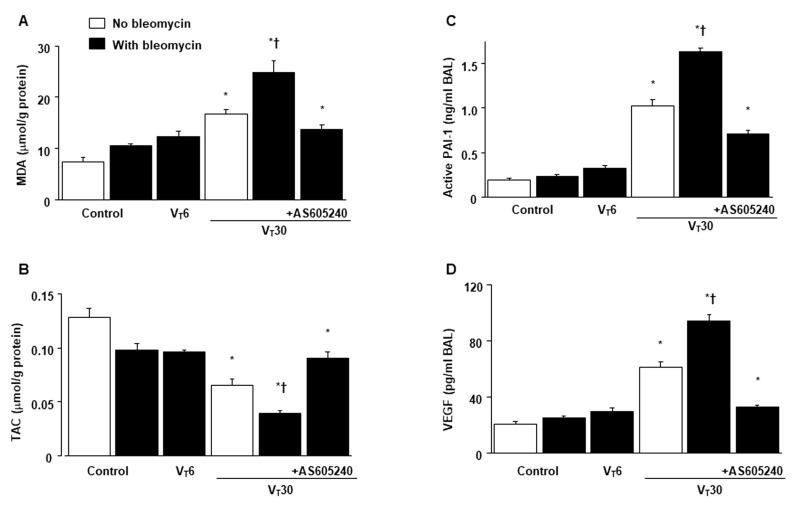
Inhibition of lung stretch-mediated oxidative stress, PAI-1, and VEGF production by AS605240. Five days after administering bleomycin, (**A**) MDA, (**B**) TAC, (**C**) PAI-1, and (**D**) VEGF secretion in BAL fluid from the lungs of nonventilated control mice and those subjected to a tidal volume at 6 or 30 mL/kg for 5 h with room air (*n* = 5 per group). AS605240 5 mg/kg was given intraperitoneally 1 h before ventilation. * *p* < 0.05 versus the nonventilated control mice with bleomycin pretreatment; † *p* < 0.05 versus all other groups. BAL = bronchoalveolar lavage; MDA = malondialdehyde; PAI-1 = plasminogen activator inhibitor-1; TAC = total antioxidant capacity; and VEGF = vascular endothelial growth factor.

**Figure 3 ijms-24-05538-f003:**
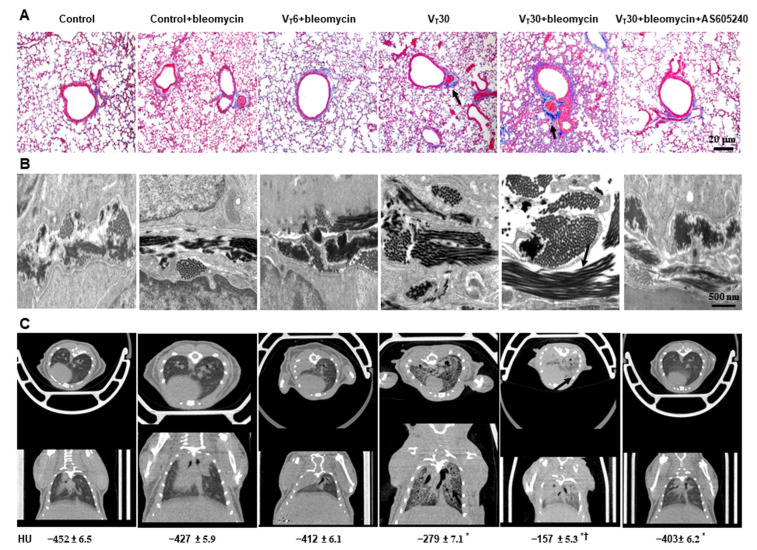
Reduction of lung stretch-induced collagen production by AS605240. (**A**) Representative micrographs (×100) with Masson’s trichrome staining of paraffin lung sections (*n* = 5 per group), (**B**) TEM micrographs (×75,000, *n* = 3 per group) of the lung sections, and (**C**) micro-computed tomography imaging (*n* = 3 per group) of lung tissue after five days of bleomycin administration from nonventilated control mice and mice ventilated at a tidal volume of 6 or 30 mL/kg for 5 h with room air. AS605240 5 mg/kg was given intraperitoneally 1 h before ventilation. Scale bars represent 20 μm or 500 nm. * *p* < 0.05 versus the nonventilated control mice with bleomycin pretreatment; † *p* < 0.05 versus all other groups. HU = Hounsfield unit; and TEM = transition electron microscope.

**Figure 4 ijms-24-05538-f004:**
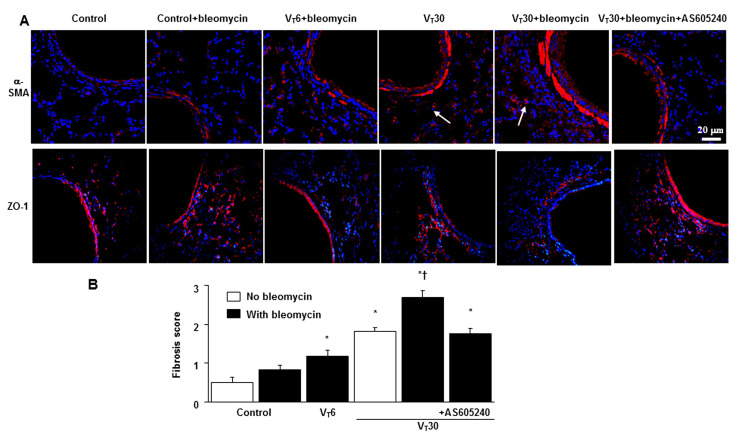
Suppression of lung stretch-induced fibrogenic markers by AS605240. (**A**) Representative photomicrographs (×400) with α-smooth muscle actin (α-SMA, red), Zonula occludens (ZO-1, red), and Hoechst (blue) immunofluorescent staining of paraffin lung sections after five days of bleomycin administration from nonventilated control mice and mice ventilated at a tidal volume of 6 or 30 mL/kg for 5 h with room air (*n* = 5 per group). Positive red staining in the lung epithelium and interstitium is identified by arrows (*n* = 5 per group). (**B**) The fibrotic scoring was quantified as the average number of 10 nonoverlapping fields in Masson’s trichrome staining of paraffin lung sections (*n* = 5 per group). AS605240 5 mg/kg was given intraperitoneally 1 h before ventilation. Scale bars represent 20 μm. * *p* < 0.05 versus the nonventilated control mice with bleomycin pretreatment; and † *p* < 0.05 versus all other groups.

**Figure 5 ijms-24-05538-f005:**
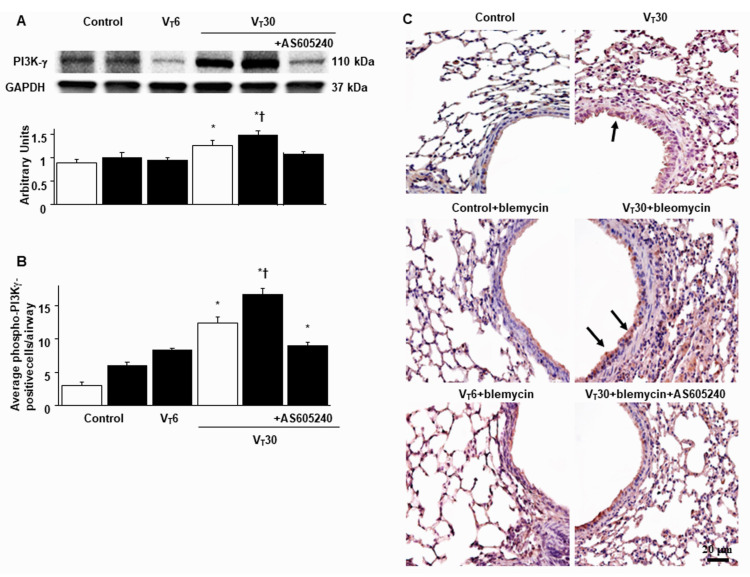
Inhibition of lung stretch-induced PI3K-γ expression by AS605240. Five days after administering bleomycin, (**A**) Western blots were performed using antibodies that recognize the expression of PI3K-γ and GAPDH in lung tissue from nonventilated control mice and mice ventilated at a tidal volume of 6 or 30 mL/kg for 5 h with room air (*n* = 5 per group). Arbitrary units are expressed as the relative PI3K-γ expression (*n* = 5 per group). (**B**,**C**) Representative micrographs (×400) with PI3K-γ staining of paraffin lung sections and quantification 5 days after administering bleomycin from nonventilated control mice and mice ventilated at a tidal volume of 6 or 30 mL/kg for 5 h with room air (*n* = 5 per group). AS605240 5 mg/kg was given intraperitoneally 1 h before ventilation. Scale bars represent 20 μm. * *p* < 0.05 versus the nonventilated control mice with bleomycin pretreatment; † *p* < 0.05 versus all other groups. GAPDH = glyceraldehyde-3-phosphate dehydrogenase; and PI3K-γ = phosphoinositide 3-kinase-γ.

**Figure 6 ijms-24-05538-f006:**
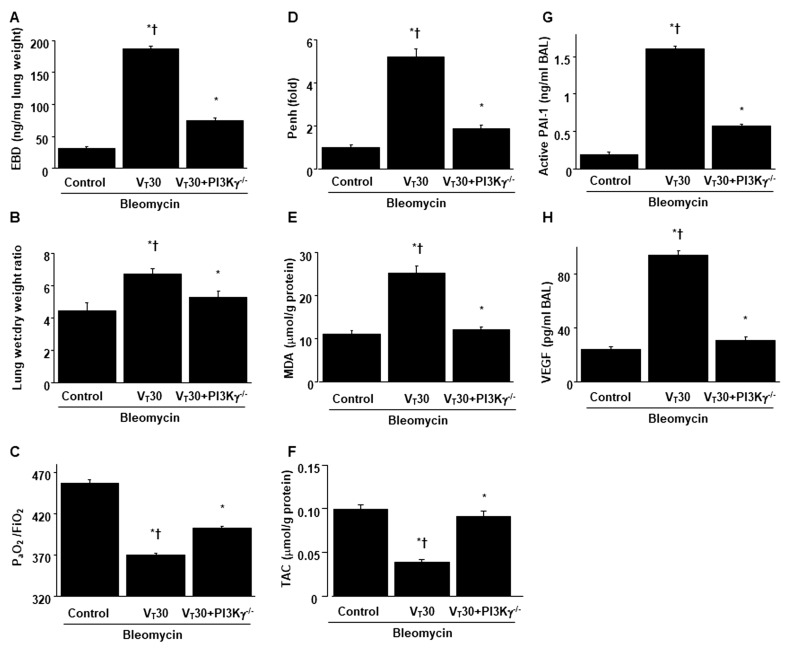
Reduction of lung stretch-induced lung inflammation in PI3K-γ-deficient mice. (**A**) Evans blue dye analysis, (**B**) lung wet-to-dry-weight ratio), (**C**) PaO_2_/FiO_2_, (**D**) enhanced pause, (**E**) MDA, (**F**) TAC, (**G**) PAI-1, and (**H**) VEGF from the lungs of nonventilated control mice and those subjected to a tidal volume of 30 mL/kg for 5 h with bleomycin administration (*n* = 5 per group). * *p* < 0.05 versus the nonventilated control mice with bleomycin; and † *p* < 0.05 versus PI3K-γ-deficient mice.

**Figure 7 ijms-24-05538-f007:**
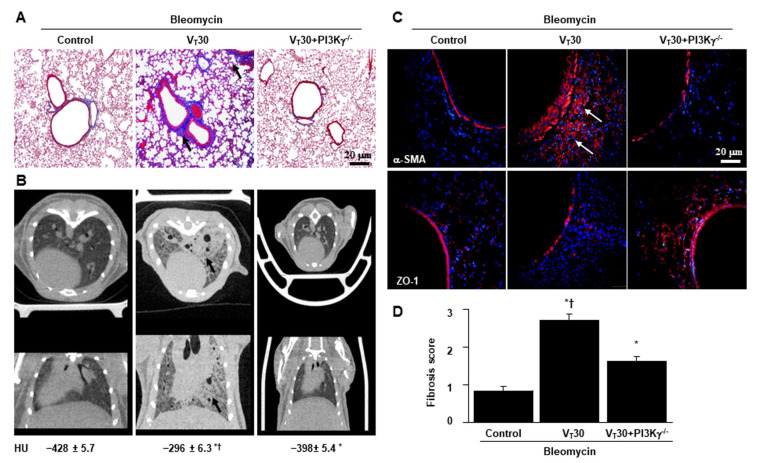
Inhibition of lung stretch-induced collagen accumulation and fibrogenic markers in PI3K-γ-deficient mice. (**A**) Representative micrographs (×100) with Masson’s trichrome staining (*n* = 5 per group), (**B**) micro-computed tomography imaging (*n* = 3 per group), and (**C**) representative photomicrographs (×400) with α-SMA (red), ZO-1 (red), and Hoechst (blue) (*n* = 5 per group) immunofluorescent staining of paraffin lung sections after five days of bleomycin administration from nonventilated control mice and mice ventilated at a tidal volume of 30 mL/kg for 5 h with room air (*n* = 5 per group). (**D**) Positive red staining in the lung epithelium and interstitium is identified by arrows (*n* = 5 per group). The fibrotic scoring was quantified as the average number of 10 nonoverlapping fields in Masson’s trichrome staining of paraffin lung sections. Scale bars represent 20 μm. * *p* < 0.05 versus the nonventilated control mice with bleomycin pretreatment; and † *p* < 0.05 versus PI3K-γ-deficient mice.

**Figure 8 ijms-24-05538-f008:**
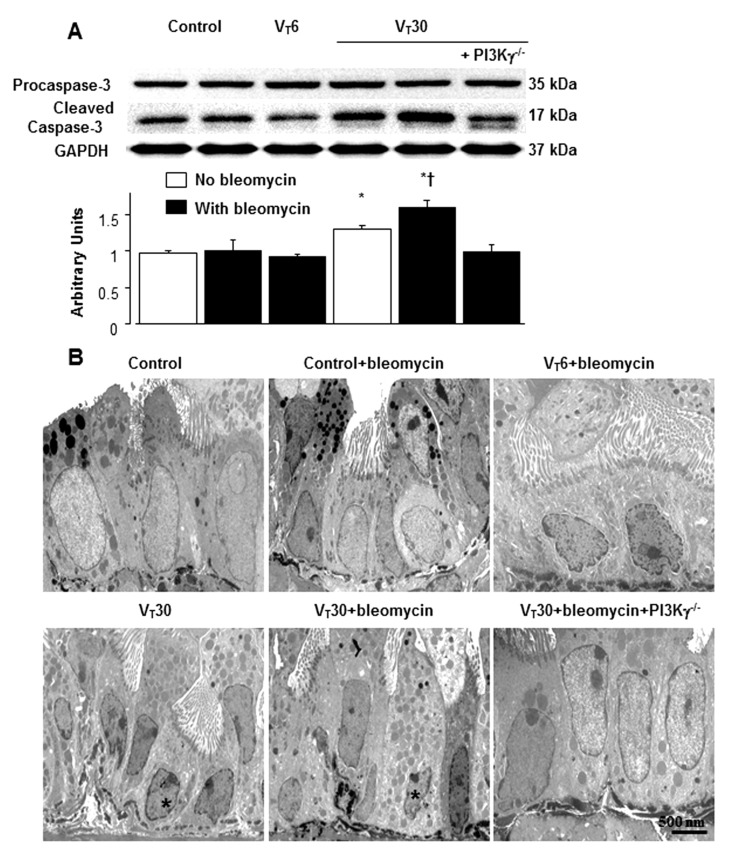
PI3K-γ homozygous knockout ameliorated lung stretch-induced caspase-3 expression and epithelial apoptosis. Five days after administering bleomycin, (**A**) Western blots were conducted using antibodies that recognize caspase-3 and GAPDH expression in lung tissue from nonventilated control mice and mice ventilated at a tidal volume of 6 or 30 mL/kg for 5 h with room air (*n* = 5 per group). Arbitrary units are expressed as the ratio of cleaved caspase-3 to GAPDH (*n* = 5 per group). (**B**) Representative TEM micrographs (×11,000, *n* = 3 per group) of the lung sections from the lungs of nonventilated control mice and mice ventilated at a tidal volume of 6 or 30 mL/kg for 5 h with bleomycin administration (*n* = 3 per group). Highly condensed and fragmented heterochromatin of epithelial cells indicates apoptosis. Apoptotic cells are identified by asterisks or arrows. Scale bars represent 500 nm. * *p* < 0.05 versus the nonventilated control mice with bleomycin; and † *p* < 0.05 versus PI3K-γ-deficient mice.

**Figure 9 ijms-24-05538-f009:**
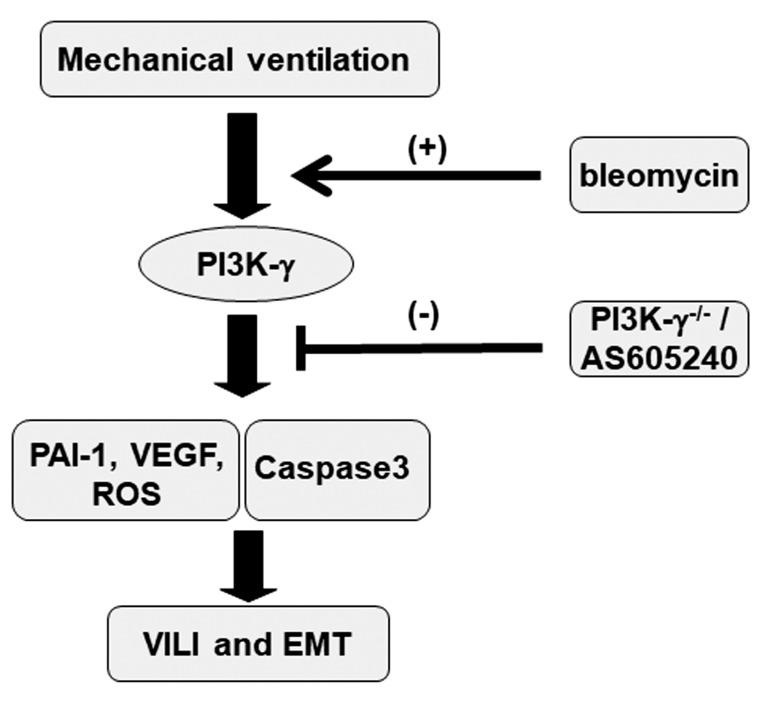
Schematic figure illustrating the signaling pathway activation with mechanical ventilation and bleomycin. Bleomycin-induced augmentation of mechanical stretch-mediated cytokine production and lung damage were attenuated in PI3K-γ-deficient mice and pharmacological inhibition with AS605240. EMT = epithelial–mesenchymal transition; ROS = reactive oxygen species; PAI-1 = plasminogen activator inhibitor-1; PI3K-γ^-/-^ = PI3K-γ-deficient mice; VEGF = vascular endothelial growth factor; and VILI = ventilator-induced lung injury.

## Data Availability

The data presented in this study are available on request from the corresponding author.

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
