# Peer review of "Attenuation of Ventilation-Enhanced Epithelial–Mesenchymal Transition through the Phosphoinositide 3-Kinase-γ in a Murine Bleomycin-Induced Acute Lung Injury Model"

_ijms, 2023, doi:10.3390/ijms24065538_

Round 1
Reviewer 1 Report
Although it lack some novelty, the manuscript is scientifically solid ant the message it is clear. Therefore, I do not have any objection to its acceptance on IJMS.
Author Response
Thanks for your comments.
Reviewer 2 Report
In this manuscript, Li-Fu Li and collaborators attempt to address a putative role of PI3K-gamma pathway during epithelial-mesenchymal transition by modulating oxidative stress and apoptosis during injury. The idea is interesting, but the main concern for this reviewer is the lack of novelty and a strong mechanism to support that.
Major comments:
1.- Role of PI3K regulating EMT and apoptosis have deeply proof in previous literature. The authors could strength the relevance and novelty of the present research study by a delineating a more detailed mechanisms that is lacking in the current form of this manuscript.
It is not clear for this reviewer what is the experimental design behind the bleomycin model, “The mice received a single dosage 415 of 0.075 units of bleomycin in 100 l of sterile normal saline (2 mg/kg, Sigma, St. Louis, 416 MO, USA) intratracheally for 5 days” Do they mean that it was a single dose and then evaluate changes at day 5 after injury?If so, during this phase, there is a peak in the inflammatory response that can be similar to ALI, it is not clear why the authors decided to add a second injury (mechanical ventilation).
2.- Following that point, the authors proponed that pharmacological inhibition of PI3K signaling by administration of AS605240 results in decreased fibrotic response; however, day 5 after bleomycin-induced fibrosis is not adequate time to evaluate fibrosis, mainly, the authors could have evaluate the inflammatory response in BAL.
3.- Trichrome staining images in Fig 3A, 5C, and 7A are showing collagen around airways, fibrotic associated collagen is present in the interstitium. Addition of images of the total lung section may be needed. The authors should clarify this discrepancy. In addition to the evaluation of fibrosis by another methods including hydroxyproline assay.
4.- It is concerning that control group (no bleomycin) shows apoptotic response measured by cleaved caspase-3 where total caspase 3 levels are missing. Confirmation of this finding should be validated by increasing/repeating sample#/experiments and adding different forms of evaluation such as caspase activity or immunofluorescence.
5.- Different cellular response, including senescence have been associated with chromatin changes (), it is not clear with a TEM analysis how chromatin modification leads to apoptosis as an only player. STORM analysis of epigenetics marks (histones modifications) - associated with apoptosis might help to clarify this point.
6.- The lung is one of the most diverse organs in terms of cellular types. It is not clear for this reviewer, which type of epithelial cells are the main focus for this study, or if the proposed idea applies to a general EMT.
Author Response
We would like to thank the reviewers for their assistance in improving our manuscript. We have updated the references and responded to reviewers as follows:
Reviewer 2
In this manuscript, Li-Fu Li and collaborators attempt to address a putative role of PI3K-gamma pathway during epithelial-mesenchymal transition by modulating oxidative stress and apoptosis during injury. The idea is interesting, but the main concern for this reviewer is the lack of novelty and a strong mechanism to support that.
Major comments:
1-1. Role of PI3K regulating EMT and apoptosis have deeply proof in previous literature. The authors could strength the relevance and novelty of the present research study by a delineating a more detailed mechanisms that is lacking in the current form of this manuscript.
Ans: In line 315-325, we added “MV is crucial to maintain adequate ventilation and gas exchange in patients with ARDS; however, MV can trigger VILI or aggravate the inflammatory response to fibrogenesis, leading to VILI-associated lung fibrosis. Though the ARDS network’s clinical trials have revealed that low-tidal-volume MV is safer than high-tidal-volume MV, the mortality of patients with ARDS has remained substantially high; the mechanisms of MV-associated lung fibrosis or post-ARDS pulmonary fibrogenesis need to be further investigated [5, 26, 27]. In this study, we demonstrated that extensive epithelial injury results in increased alveolar capillary permeability and inflammatory parameters in our two-hit animal model. MV-augmented bleomycin-induced pulmonary fibrogenesis and epithelial apoptosis were attenuated in PI3K-γ-deficient mice and pharmacological inhibition of PI3K-γ activity through AS605240.”
1-2. It is not clear for this reviewer what is the experimental design behind the bleomycin model, “The mice received a single dosage 415 of 0.075 units of bleomycin in 100 l of sterile normal saline (2 mg/kg, Sigma, St. Louis, 416 MO, USA) intratracheally for 5 days” Do they mean that it was a single dose and then evaluate changes at day 5 after injury?If so, during this phase, there is a peak in the inflammatory response that can be similar to ALI, it is not clear why the authors decided to add a second injury (mechanical ventilation).
Ans. In line 297-304, we make a revision of description about the purpose of our study as
- Fibroblast proliferation and ECM accumulation are started 4-14 days after bleomycin challenge [23, 24]. Furthermore, Tager et al. demonstrated that the angiogenic and fibroblast chemotactic activities generated in the mouse lung after 5 days from a bolus of bleomycin administration were substantially upregulated [24]. Further, we have shown that there were significant increases of fibroblast accumulation and fibrogenic markers in this animal model using MV after 5 days from bleomycin administration [1, 2]. Five days of bleomycin administration was thus used in our study to focus on the major target involved in the early phase of pulmonary fibrosis after ALI.
- In line 315-325, we added “MV is crucial to maintain adequate ventilation and gas exchange in patients with ARDS; however, MV can trigger VILI or aggravate the inflammatory response to fibrogenesis, leading to VILI-associated lung fibrosis. Though the ARDS network’s clinical trials have revealed that low-tidal-volume MV is safer than high-tidal-volume MV, the mortality of patients with ARDS has remained substantially high; the mechanisms of MV-associated lung fibrosis or post-ARDS pulmonary fibrogenesis need to be further investigated [5, 26, 27]. In this study, we demonstrated that extensive epithelial injury results in increased alveolar capillary permeability and inflammatory parameters in our two-hit animal model. MV-augmented bleomycin-induced pulmonary fibrogenesis and epithelial apoptosis were attenuated in PI3K-γ-deficient mice and pharmacological inhibition of PI3K-γ activity through AS605240.”
- Following that point, the authors proponed that pharmacological inhibition of PI3K signaling by administration of AS605240 results in decreased fibrotic response; however, day 5 after bleomycin-induced fibrosis is not adequate time to evaluate fibrosis, mainly, the authors could have evaluate the inflammatory response in BAL.
Ans.
In line 298-304, we added the description as “Tager et al. demonstrated that the angiogenic and fibroblast chemotactic activities generated in the mouse lung after 5 days from a bolus of bleomycin administration were substantially up-regulated [24]. Further, we have shown that there were significant increases of fibroblast accumulation and fibrogenic markers in this animal model using MV after 5 days from bleomycin administration [1, 2].
In line 226-236, we added “In the course of ARDS, lung inflammation and fibrogenesis to some degree interact [2]. We observed that neutrophil counts in the BAL fluid increased in mice subjected to VT at 30 ml/kg compared with those subjected to VT at 6 ml/kg and the control mice. Significantly, PI3K-γ knockout and pharmacologic inhibition with AS605240 substantially reduced neutrophil infiltration (neutrophil counts: nonventilated control wild-type mice without bleomycin = 1.0±0.1, nonventilated control wild-type mice with bleomycin = 2.1±0.1, VT 6 ml/kg wild-type mice with bleomycin = 5.7±0.3*, VT 30 ml/kg wild-type mice without bleomycin = 24.1±1.8*, VT 30 ml/kg wild-type mice with bleomycin = 42.5±2.6*, VT 30 ml/kg PI3K-deficient mice with bleomycin = 20.7±1.4*, VT 30 ml/kg wild-type mice after AS605240 with bleomycin = 28.3±2.9* x 104/ml BAL, *P < 0.05 versus control (Figure S3).”
3.- Trichrome staining images in Fig 3A and 7A are showing collagen around airways, fibrotic associated collagen is present in the interstitium. Addition of images of the total lung section may be needed. The authors should clarify this discrepancy. In addition to the evaluation of fibrosis by another methods including hydroxyproline assay.
Ans.
- In line 341-345, we added “ALI is characterized by impaired gas exchange of alveoli and airways, which includes terminal and respiratory bronchioles. Therefore, we explored high-tidal-volume MV-induced peribronchiolar and parenchymal collagen accumulation in murine bronchial epithelium besides the alveolar injury found in our (Figures 2A and 7A) and others’ studies in the mice [29]. “
- We have added a supplementary Figure S1 to demonstrate total lung image in our study.
- We didn’t perform hydroxyproline.analysis and discuss it in the limitation. In line 443-448, we added “Zhang R et al. demonstrated that MV can induce the lung fibrotic changes in an animal model of ARDS-associated lung fibrosis, as evidenced by enhanced levels of hydroxyproline in lung tissues [44]. Hydroxyproline, a crucial component of the protein collagen as an early biochemical marker contributing to collagen synthesis, can serve as a supportive assay in addition to the Masson's trichrome staining used in our study.”
4.- It is concerning that control group (no bleomycin) shows apoptotic response measured by cleaved caspase-3 where total caspase 3 levels are missing. Confirmation of this finding should be validated by increasing/repeating sample#/experiments and adding different forms of evaluation such as caspase activity or immunofluorescence.
Ans: We added total caspase data in Figure 8.
In line 261-264, we added “The substantial increase in the expression of cleaved caspase-3 (active form) but decrease in the expression of non-cleaved caspase-3 was observed in the mice treated with bleomycin and high-tidal-volume MV compared with the other MV treatment groups and the nonventilated control mice (Figure 8A).”
5.- Different cellular response, including senescence have been associated with chromatin changes (), it is not clear with a TEM analysis how chromatin modification leads to apoptosis as an only player. STORM analysis of epigenetics marks (histones modifications) - associated with apoptosis might help to clarify this point.
Ans: In line 436-443, we add the description as below in the limitation of our study. ”Histone modification has been demonstrated to be involved in the higher-order chromatin structure alterations in cellular apoptosis [41]. Although our pathologic findings of epithelial apoptosis revealed characteristic nuclear condensation and fragmented heterochromatin of epithelial cells, different cellular responses including senescence have been associated with chromatin changes [42]. Super-resolution microscopy (e.g., stochastic optical reconstruction microscopy [STORM]) may help to detect the epigenetic marks to delineate the pattern of histone medication [43].”
6.- The lung is one of the most diverse organs in terms of cellular types. It is not clear for this reviewer, which type of epithelial cells are the main focus for this study, or if the proposed idea applies to a general EMT.
Ans: In line 424-429, we add the description as “In the current study, bronchial epithelial cells are our main focus to investigate. Immunohistochemistry was used to further define the cell types involved in the lung stretch-induced fibrogenesis and verify the effects of AS605240 on PI3K-γ activation in VILI (Figures 5B, 5C). Previous in vitro study demonstrated that primary human bronchial epithelial cells, in addition to type 2 alveolar epithelial cells, stimulated by TGF-β1 can initiate EMT process through a Smad 2/3-mediated pathway [39].
Reviewer 3 Report
The manuscript "Attenuation of ventilation-enhanced epithelial-mesenchymal transition through the phosphoinositide 3-kinase-r in a murine bleomycin-induced acute lung injury model" provides new insights into MV, EMT, and ALI . This report is potentially interesting, but the manuscript can be improved according to the following suggestions:
1. In figure 3A and 7A, control+bleo should induce inflammatory cell infiltration. But I cannot see a typical image here. Besides, all the images should be amplificated for a more clear version.
2. In figure 4A, the VT6+bleo treated group showed a lower a-SMA and ZO-1 expression when compare to control+bleo. But, in HE and other index, this group showed a more severe injury when compared to control+bleo group. The authors should explain the reason for the differences.
3. The author should add groups that treat with AS605240 without VT, and PI3Kγ KO mice treat with bleo in the absence of VT. These groups will make the PI3Ky pathway specifically target VT but not bleo. The injury caused by VT and bleo is different and should be distinguished
Author Response
We would like to thank the reviewers for their assistance in improving our manuscript. We have updated the references and responded to reviewers as follows:
Reviewer 3
The manuscript "Attenuation of ventilation-enhanced epithelial-mesenchymal transition through the phosphoinositide 3-kinase-r in a murine bleomycin-induced acute lung injury model" provides new insights into MV, EMT, and ALI . This report is potentially interesting, but the manuscript can be improved according to the following suggestions:
- In figure 3A and 7A, control+bleo should induce inflammatory cell infiltration. But I cannot see a typical image here. Besides, all the images should be amplificated for a more clear version.
Ans: We have corrected the resolution of Figures 3A and 7A and added a amplificated figure (Figure S2 ) for a more clear version.
- In figure 4A, the VT6+bleo treated group showed a lower a-SMA and ZO-1 expression when compare to control+bleo. But, in HE and other index, this group showed a more severe injury when compared to control+bleo group. The authors should explain the reason for the differences.
Ans: We have corrected the resolution of Figures 4A VT6+bleo treated group.
- The author should add groups that treat with AS605240 without VT, and PI3Kγ KO mice treat with bleo in the absence of VT. These groups will make the PI3Ky pathway specifically target VT but not bleo. The injury caused by VT and bleo is different and should be distinguished.
Ans. In line 268-271, we added the description as “Because no statistically significant differences were observed between the wild-type and PI3k-γ-deficient nonventilated control mice or wild-type and AS605240 treatment nonventilated control mice, both with and without bleomycin, the data are not presented in this paper.”
Round 2
Reviewer 2 Report
No further comments